# An oligogenic architecture underlying ecological and reproductive divergence in sympatric populations

Dušica Briševac, Carolina M Peralta, Tobias S Kaiser*

Max Planck Research Group Biological Clocks, Max Planck Institute for Evolutionary Biology, Ploen, Germany

**Abstract** The evolutionary trajectories and genetic architectures underlying ecological divergence with gene flow are poorly understood. Sympatric timing types of the intertidal insect *Clunio marinus* (Diptera) from Roscoff (France) differ in lunar reproductive timing. One type reproduces at full moon, the other at new moon, controlled by a circalunar clock of yet unknown molecular nature. Lunar reproductive timing is a magic trait for a sympatric speciation process, as it is both ecologically relevant and entails assortative mating. Here, we show that the difference in reproductive timing is controlled by at least four quantitative trait loci (QTL) on three different chromosomes. They are partly associated with complex inversions, but differentiation of the inversion haplotypes cannot explain the different phenotypes. The most differentiated locus in the entire genome, with QTL support, is the *period* locus, implying that this gene could not only be involved in circadian timing but also in lunar timing. Our data indicate that magic traits can be based on an oligogenic architecture and can be maintained by selection on several unlinked loci.

## Editor's evaluation

This important article provides solid evidence for an apparent oligogenic architecture of an ecologically relevant trait, the circalunar reproduction of marine midges, which contributes to assortative mating, is likely under divergent selection, and supports reproductive isolation in sympatry.

*For correspondence:
kaiser@evolbio.mpg.de

Competing interest: The authors declare that no competing interests exist.

## Introduction

Evolutionary biology has seen a long debate on whether speciation can only happen with geographic isolation (*allopatry*) or also with full range overlap (*sympatry*) *Mayr, 1947*; *Smith, 1966*; *Via, 2001*; *Foote, 2018*; *Coyne and Orr, 2004*. Today, allopatric and sympatric speciation are viewed as the two extremes on a multifaceted continuum of speciation with gene flow, for which there is growing evidence *Foote, 2018*; *Richards et al., 2019*. Speciation with gene flow is generally assumed to start with divergent ecological selection, but for speciation to complete, additional components of reproductive isolation must come into play *Smadja and Butlin, 2011*; *Seehausen et al., 2014*. Traits underlying ecological divergence and other components of reproductive isolation must be genetically coupled, as otherwise they cannot withstand the homogenizing effects of gene flow and recombination *Smadja and Butlin, 2011*; *Felsenstein, 1981*. Such coupling can be achieved via pleiotropy, magic traits, or genetic linkage between respective genes. Pleiotropy describes the situation where 'one allele affects two or more traits' *Barton et al., 2007* contributing to reproductive isolation. Here, the coupling can be achieved via a single genetic locus. A 'magic trait' *Gavrilets, 2004* in turn is a single trait which affects several components of reproductive isolation at the same time *Smadja and Butlin, 2011*; *Servedio et al., 2011*. It is therefore also referred to as a 'multiple-effect trait' *Smadja*

*and Butlin, 2011.* The concept does per se not include an assumption on the genetic basis of the trait, and should therefore not be equated with pleiotropy. Classically, magic traits are thought to affect ecological divergence and assortative mating at the same time. Of particular interest to our study is the scenario where divergent ecological adaptations lead to differences in reproductive timing, that is *allochrony Taylor and Friesen, 2017* or *isolation by time Hendry and Day, 2005.* One such example is found in the apple maggot *Rhagoletis Filchak et al., 2000*; *Doellman et al., 2018.* Finally, genetic linkage imposed by genomic regions of low or suppressed recombination, such as chromosomal inversions, can entail a coupling of ecological and reproductive traits even if they are controlled by different sets of genes *Butlin, 2005.* As an example, a chromosomal inversion in the monkey flower *Mimulus guttatus* was found to contribute to ecological adaptation and reproductive isolation *Lowry and Willis, 2010.* Other examples of inversions associated with ecologically relevant traits are found in *Littorina* snails *Koch et al., 2021*; *Faria et al., 2019* and *Heliconius* butterflies *Joron et al., 2011.* In this study, we assessed if magic traits or genetic linkage play a role in sympatric population divergence in the marine midge *Clunio marinus* (Diptera: Chironomidae).

*C. marinus* is found in the intertidal zone of the European Atlantic coast. Larvae live at the lower levels of the intertidal, which are almost permanently submerged. For successful reproduction, the adults need these regions to be exposed by the low tide. The lowest tides predictably recur just after new moon and full moon. Therefore, *C. marinus* adults emerge only during full moon or new moon low tides, reproduce immediately and die a few hours later in the rising tide. This life cycle adaptation is based on a circalunar clock, that is an endogenous time-keeping mechanism which synchronizes development and maturation with lunar phase. Circalunar clocks are common in marine organisms, but their molecular basis is still unknown *Kaiser and Neumann, 2021.* Additionally, a circadian clock ensures that *C. marinus* adults only emerge during the low tide *Neumann, 1967.* As the pattern and amplitude of the tides differ dramatically along the coastline, *C. marinus* populations from different geographic sites show various genetic adaptations in circadian and circalunar timing *Neumann, 1967*; *Kaiser, 2014*; *Kaiser et al., 2011.*

Notably, low tide water levels follow a bimodal distribution across a lunar month, with minima at both full moon and new moon. These minima are ecologically equally suitable for *C. marinus'* reproduction, representing different *timing niches* that are occupied by different *timing types* (for details and definitions see *Kaiser et al., 2021*). Some timing types of *C. marinus* use both niches ('semi-lunar rhythm'; SL type), but there are also dedicated full moon (FM) or new moon (NM) timing types, which occupy only one timing niche *Kaiser et al., 2021.* As water levels outside the minima are much less suitable for successful reproduction, we can expect divergent selection toward either full moon or new moon emergence. This divergence into FM and NM timing types represents a magic trait, as it automatically entails assortative mating. Additionally, hybrids between timing types are expected to emerge at intermediate times with respect to their parents *Neumann, 1967*; *Kaiser et al., 2011*, that is during neap tide high tides. As this tidal situation is unfavorable for *C. marinus'* reproduction, we can expect selection against hybrids, a form of an ecologically imposed postzygotic incompatibility. We recently discovered that in Roscoff (France) FM and NM timing types occur in sympatry, largely separated by reproductive timing, but still connected by gene flow *Kaiser et al., 2021.* Here, we present evidence for polymorphic chromosomal inversions in all chromosome arms of the sympatric FM and NM types. Quantitative trait loci (QTL) mapping for divergence in lunar reproductive timing identifies four unlinked and additive QTL in different chromosomes and inversions. Individual loci inside the inversions are more differentiated than the inversions, suggesting that the inversions themselves are not directly required to maintain linkage disequilibrium between adaptive loci and that ecological divergence is kept up by permanent selection on unlinked loci on different chromosomes.

## Results

### Genome sequencing confirms limited genetic differentiation between the sympatric FM and NM types

In order to gain insights into the loci and processes underlying sympatric divergence in reproductive timing, we sequenced 48 individual genomes for the FM and NM timing types (24 individuals each, defined by their lunar timing phenotype; genome size: 79.4 Mbp; average read coverage: 20 x). The resulting set of 721,000 genetic variants (608,599 SNPs; 112,401 small indels) indicated limited

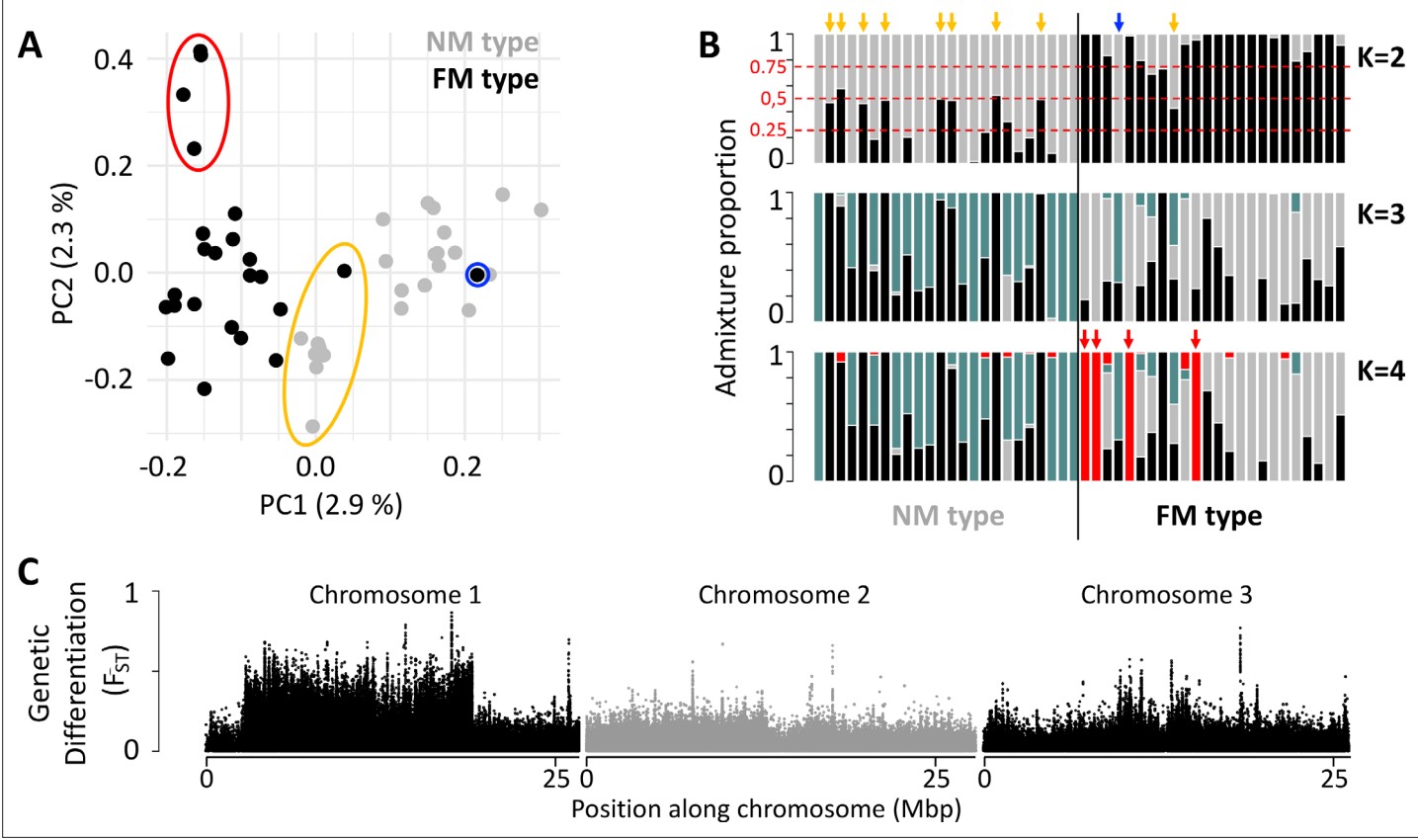

**Figure 1.** Population structure of the FM and NM types in Roscoff based on 721,000 genetic variants. (**A, B**) Principal component analysis (PCA; **A**) and admixture analysis (**B**) identify one migrant in time (blue; pure NM genotype caught at full moon), many potential F1 hybrids (yellow) and four individuals in the FM strain that appear genetically distict from all other samples (red). (**C**) Global genetic differentiation is limited, but there is a block of strong differentiation on chromosome 1.

genetic differentiation between the FM and NM types (*Figure 1*; weighted genome-wide $F_{ST}$ = 0.028). Principal component analysis (PCA) and ADMIXTURE identify one individual as a *migrant in time*, which was caught at full moon among FM type individuals, but genetically clearly is of NM type (*Figure 1A and B*; blue circle and arrow). Many other individuals, particularly in the NM type, show ADMIXTURE fractions close to 0.5 or 0.25, suggesting they are F1 hybrids or backcrosses (*Figure 1A and B*; yellow circle and arrows). Finally, there are four FM individuals, which are genetically distinct along principal component 2 and in ADMIXTURE at K=4 (*Figure 1A and B*; red circle and arrows). These might constitute either a sub-lineage of the FM type or migrants from a different geographic location. Taken together, full genome resequencing confirms that there is considerable hybridization of sympatric timing types and overall low genome-wide differentiation.

## A differentiated chromosomal inversion system on chromosome 1

Plotting genetic differentiation of the phenotypically defined FM and NM types along the genome (*Figure 1C*) revealed a region of elevated $F_{ST}$ on the telocentric chromosome 1 (*Figure 2A*). This region coincides with a block of long-range linkage-disequilibrium (LD) in the FM type (*Figure 2B*; *Figure 2—figure supplement 1*). Structural variant (SV) calling based on additional long-read sequencing data supports that this block represents a chromosomal inversion polymorphic in both timing types, hereby called In(1 a) (*Figure 2C*; full dataset in *Supplementary file 1*). Notably, the NM type shows a smaller LD block (*Figure 2B*; *Figure 2—figure supplement 1*), suggesting that a second structural variant occurs within the limits of the detected large chromosomal inversion. While this second variant was not picked up in SV calling, genetic linkage information obtained from crosses (described in detail below) indicates that this is a second inversion, named In(1b) (see double inverted marker order in Figure 5A and B).

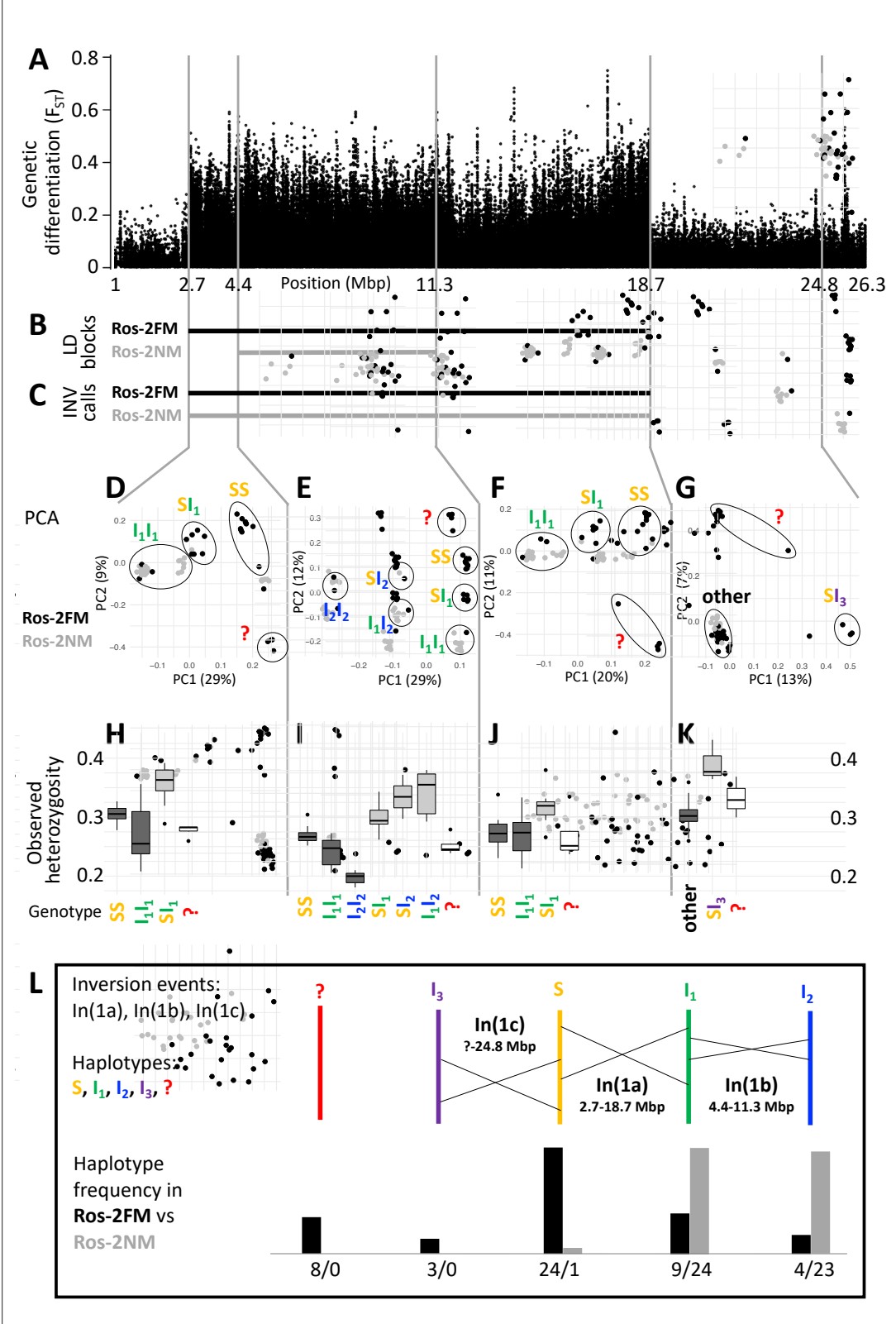

**Figure 2.** A complex inversion system on chromosome 1. (**A**) Chromosome 1 harbors a block of markedly elevated genetic differentiation. (**B**) This genomic block coincides with two windows of elevated long-range LD in the FM and NM strains. (**C**) Structural variant (SV) calling from long read sequencing data supports that the larger LD block is due to an inversion. (**D–G**) Principal component analysis (PCA) for chromosomal sub-windows corresponding to suggested inversions separates the individuals into clusters corresponding to standard haplotype homozygotes (SS), inversion

*Figure 2 continued on next page*

*Figure 2 continued*

homozygotes ($I_1I_1$, $I_2I_2$) and inversion heterozygotes ($SI_1$, $SI_2$, $SI_3$, $I_1I_2$). The four individuals which were already found distinct in whole genome analysis (red question mark) cannot be assessed. (**H–K**) Observed heterozygosity is clearly elevated in inversion heterozygotes, underpinning substantial genetic differentiation between the inversions. (**L**) A schematic overview of the sequence of inversion events and the resulting haplotypes. The frequencies of the haplotypes differ markedly between the FM and NM types.

The online version of this article includes the following figure supplement(s) for figure 2:

**Figure supplement 1.** Long-range linkage disequilibrium along chromosome 1.

**Figure supplement 2.** Local admixture plots for the genomic windows corresponding to different inversions.

**Figure supplement 3.** Genetic differentiation between homozygotes of the standard haplotype (SS) and inversion haplotypes ($I_1I_1$, $I_2I_2$).

**Figure supplement 4.** Windowed PCA along chromosome 1, part 1 (500kb sliding windows with 100 kb steps).

**Figure supplement 5.** Windowed PCA along chromosome 1, part 2 (500kb sliding windows with 100 kb steps).

**Figure supplement 6.** Windowed PCA along chromosome 1, part 3 (500kb sliding windows with 100 kb steps).

**Figure supplement 7.** Windowed PCA along chromosome 1, part 4 (500kb sliding windows with 100 kb steps).

**Figure supplement 8.** Windowed PCA along chromosome 1, part 5 (500kb sliding windows with 100 kb steps).

**Figure supplement 9.** Windowed PCA along chromosome 1, part 6 (500kb sliding windows with 100 kb steps).

**Figure supplement 10.** Windowed PCA along chromosome 1, part 7 (500kb sliding windows with 100 kb steps).

**Figure supplement 11.** Windowed PCA along chromosome 1, part 8 (500kb sliding windows with 100 kb steps).

**Figure supplement 12.** Windowed PCA along chromosome 1, part 9 (500kb sliding windows with 100 kb steps).

**Figure supplement 13.** Windowed PCA along chromosome 1, part 10 (500kb sliding windows with 100 kb steps).

**Figure supplement 14.** Windowed PCA along chromosome 1, part 11 (500kb sliding windows with 100 kb steps).

**Figure supplement 15.** Windowed PCA along chromosome 1, part 12 (500kb sliding windows with 100 kb steps).

**Figure supplement 16.** Windowed PCA along chromosome 1, part 13 (500kb sliding windows with 100 kb steps).

**Figure supplement 17.** Windowed PCA along chromosome 1, part 14 (500kb sliding windows with 100 kb steps).

**Figure supplement 18.** Windowed PCA along chromosome 1, part 15 (500kb sliding windows with 100 kb steps).

**Figure supplement 19.** Windowed PCA along chromosome 1, part 16 (500kb sliding windows with 100 kb steps).

**Figure supplement 20.** Windowed PCA along chromosome 1, part 17 (500kb sliding windows with 100 kb steps).

**Figure supplement 21.** Windowed PCA along chromosome 1, part 18 (500kb sliding windows with 100 kb steps).

**Figure supplement 22.** Windowed analysis for observed heterozygosity (**A**) and Admixture (**B**) along chromosome 1.

For a more detailed analysis of this inversion system, we subdivided it into three windows, based on the inversion coordinates suggested by long-range LD (*Supplementary file 2*; *Supplementary file 3*). Two windows correspond to those parts of the large inversion that do not overlap with the small inversion (roughly 2.7–4.4 Mbp and 11.3–18.7 Mbp; *Supplementary file 2*; *Figure 2A–C*). The central window corresponds to the overlap of both inversions (roughly 4.4–11.3 Mbp; *Supplementary file 2*; *Figure 2A–C*). In a principal component analysis (PCA) on the genetic variants in these windows (*Supplementary file 3*), the four FM individuals that were already identified as genetically distinct (*Figure 1*) always cluster separately and cannot be assessed (*Figure 2D-G*, red question mark). The remaining 44 individuals are split into three genotype classes when PCA is performed on the non-overlapping regions of the large inversion (*Figure 2D and F*). These classes correspond to individuals homozygous for the standard haplotype (SS), individuals homozygous for an inverted haplotype ($I_1I_1$), and heterozygotes for the inversion ($SI_1$). Genotype assignments are confirmed by local admixture analysis (*Figure 2—figure supplement 2*), as well as heterozygosity (*Figure 2H–K*). Local observed heterozygosity is clearly elevated in individuals heterozygous for the inversion (*Figure 2H and J*), suggesting there are polymorphisms specific to the S or $I_1$ haplotypes. This is corroborated by the patterns of genetic differentiation between SS and $I_1I_1$ homozygotes, which show $F_{ST}$ values of up to 1 (*Figure 2—figure supplement 3*). For the homozygous genotypes, the haplotype which has the higher observed heterozygosity in homozygotes is considered the ancestral standard haplotype S (see *Figure 2H–J*).

In the region where both inversions overlap (*Figure 2E*), the 44 individuals separate into six genotype clusters, corresponding to three clusters of inversion haplotype homozygotes (SS, $I_1I_1$, $I_2I_2$) and three clusters of heterozygotes ($SI_1$, $SI_2$, $I_1I_2$). Individuals that are $I_1I_1$ genotypes in the 2.7–4.4 Mbp and

11.3–18.7 Mbp windows (*Figure 2D and F*) now separate into the $I_1I_1$, $I_1I_2$ and $I_2I_2$ clusters (*Figure 2E*). This indicates that the small inversion leading to haplotype $I_2$ happened in the already inverted haplotype $I_1$ of the large inversion (compare *Figure 2L*). This is backed up by genetic linkage data (Figure 5A and B) and consistent with $I_2I_2$ homozygotes showing even lower heterozygosity than $I_1I_1$ homozygotes (*Figure 2I*). Again, heterozygosity is clearly elevated in the inversion heterozygotes (*Figure 2I*). A sliding-window PCA analysis along the chromosomes (500 kb windows, 100 kb steps; *Figure 2—figure supplements 4–21*) confirms these patterns of segregation and the approximate genomic positions of inversion breakpoints.

Sliding-window PCA also indicated that there might be a third inversion, which ends at 24.8 Mbp (*Figure 2A and G*; *Figure 2—figure supplements 4–21*). Its start overlaps with the differentiated inversion system and is therefore hard to identify, but according to observed heterozygosity along the chromosome, it might lie at around 9 Mbp (*Figure 2—figure supplement 22A*). This inversion is only clearly detected in three $SI_3$ heterozygotes (*Figure 2G and K*). These three individuals are found in the SS cluster in the other genomic windows, implying that the inversion leading to haplotype $I_3$ happened in the standard haplotype S (*Figure 2L*).

One peculiarity deserves further investigation: In the window from 2.7 to 4.4 Mbp some individuals of the $I_1I_1$ genotype are found close to the $SI_1$ heterozygotes. However, the pattern in which these individuals segregate in the 4.4–11.3 Mbp window unequivocally indicates that these must be $I_1I_1$ homozygotes. They also show much lower heterozygosity than the $SI_1$ heterozygotes. Their peculiar clustering may indicate a certain degree of gene conversion between inversion haplotypes or a complex demographic history.

Taken together, our data support three inversion events, which we name In(1 a), In(1b) and In(1 c), leading to four distinct haplotypes of chromosome 1 (S, $I_1$, $I_2$ and $I_3$; *Figure 2L*). From the PCA-derived genotypes (*Figure 2D–G*), we can infer the frequency of each haplotype (*Figure 2L*). There is marked genetic differentiation between the sympatric timing types. The FM type carries all haplotypes but is dominated by the S haplotype, while the NM type almost exclusively carries the inverted $I_1$ and $I_2$ haplotypes. The distribution of haplotypes also explains the observed patterns of LD. As the NM type is segregating almost exclusively for $I_1$ and $I_2$, it shows recombination suppression only over the length of $I_2$. The FM type is segregating for the S haplotype vs both $I_1$ and $I_2$, and thus shows recombination suppression along $I_1$.

## Non-differentiated inversions on chromosomes 2 and 3

The analysis of long-range LD also identifies one putative chromosomal inversion on each arm of the metacentric chromosomes 2 and 3 (*Figure 3B and C*; *Figure 3—figure supplement 1*). We call them In(2 L), In(2R), In(3 L) and In(3 R). In(2 L) is confirmed by inverted marker order in linkage mapping (Figure 5A and B) and In(2 R) is confirmed by SV calling (*Supplementary file 1*). As above, windowed PCA along the chromosomes confirms the approximate inversion breakpoints and allows to genotype the individuals for the inversion haplotypes (*Figure 3D–G*). Heterozygosity is clearly elevated in inversion heterozygotes (SI; *Figure 3H–K*) and lowered in the inversion homozygotes (II; *Figure 3H–K*). Notably, inversion homozygotes are absent for two of these inversions (*Figure 3D and F*) and very rare for the other two (*Figure 3E and G*), suggesting these inversions might impose some fitness constraints. These four inversions are only weakly differentiated between populations (*Figure 3L–O*). Only In(2 L) shows up as a block of mildly elevated genetic differentiation between the strains (*Figure 3A*).

## Crosses between the FM and NM types indicate that lunar reproductive timing is heritable and controlled by at least four QTL

Next, we tested for a genetic basis of lunar reproductive timing by performing crossing experiments between the FM and NM types (*Figure 4*). F1 hybrids emerge at an intermediate time point between the FM and NM type emergence times, with a slight shift toward FM emergence (*Figure 4C*). In the F2 generation, the phenotype distribution is spread out, but does not fully segregate into parental and F1 phenotype classes (*Figure 4D*). This indicates that the difference in lunar reproductive timing is controlled by more than one genetic locus. From the crossing experiment, we picked a set of several F2 families that together comprised 158 individuals, which all go back to a single parental pair – a FM type mother and a NM type father. Because of the limited genetic differentiation between the timing

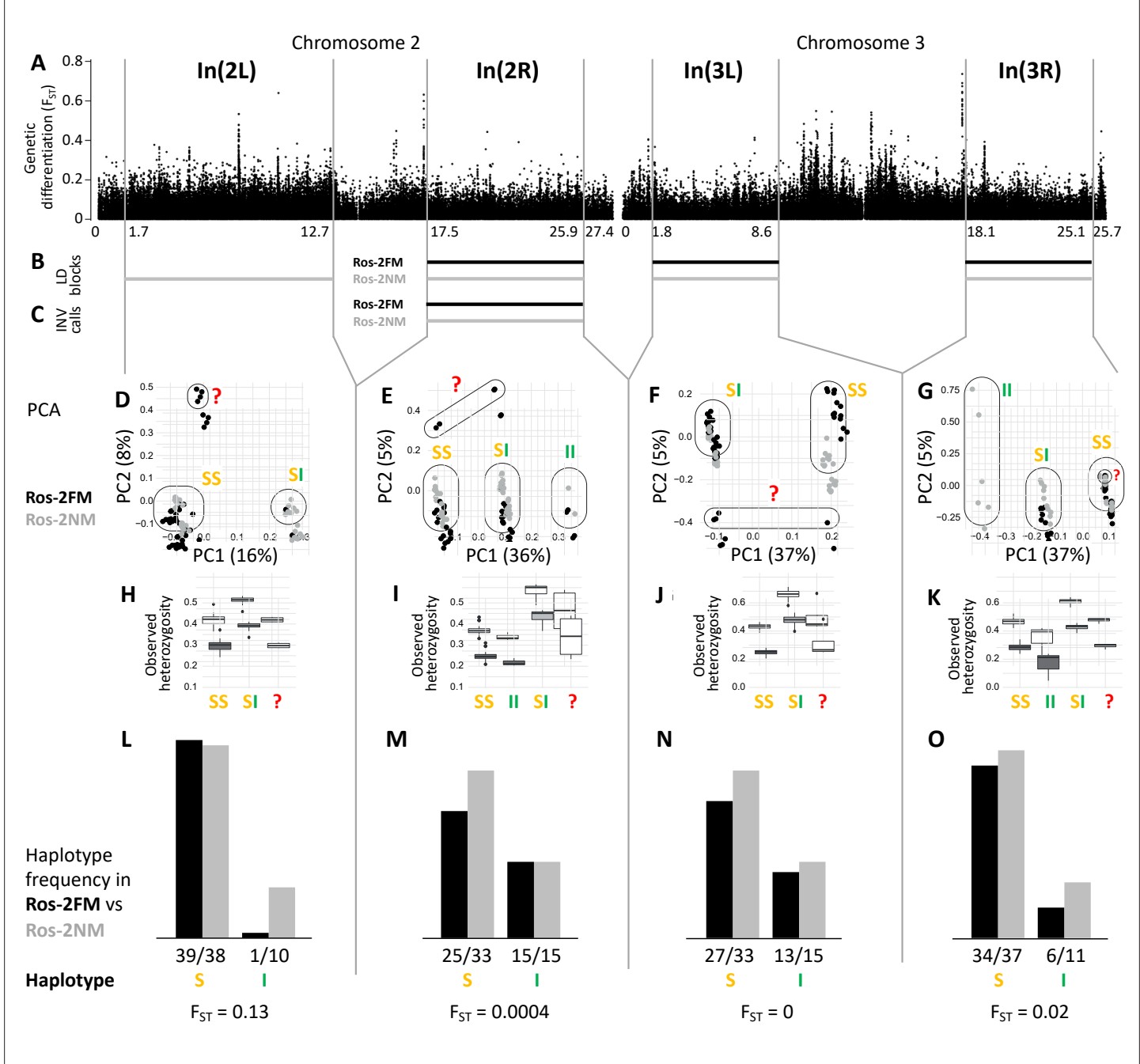

**Figure 3.** Inversions in chromosomes 2 and 3. (**A**) Chromosome arm 2L harbors a block of mildly elevated genetic differentiation. (**B**) Several blocks of long-range linkage disequilibrium (LD) point to additional inversions in the FM and NM strains, one on each chromosome arm. (**C**) Structural variant (SV) calling from long read sequencing data supports the inversion on chromosome arm 2 R. (**D–G**) Principal component analysis (PCA) for chromosomal sub-windows corresponding to suggested inversions separates the individuals into clusters corresponding to standard haplotype homozygotes (SS), inversion homozygotes (II) and inversion heterozygotes (SI). The four individuals which were already found distinct in whole genome analysis (red question mark) cannot be assessed. (**H–K**) Observed heterozygosity is clearly elevated in inversion heterozygotes, underpinning substantial genetic differentiation between the inversions. (**L–O**) The frequencies of the haplotypes do not differ much between the FM and NM types.

The online version of this article includes the following figure supplement(s) for figure 3:

**Figure supplement 1.** Long-range linkage disequilibrium along chromosomes 2 and 3.

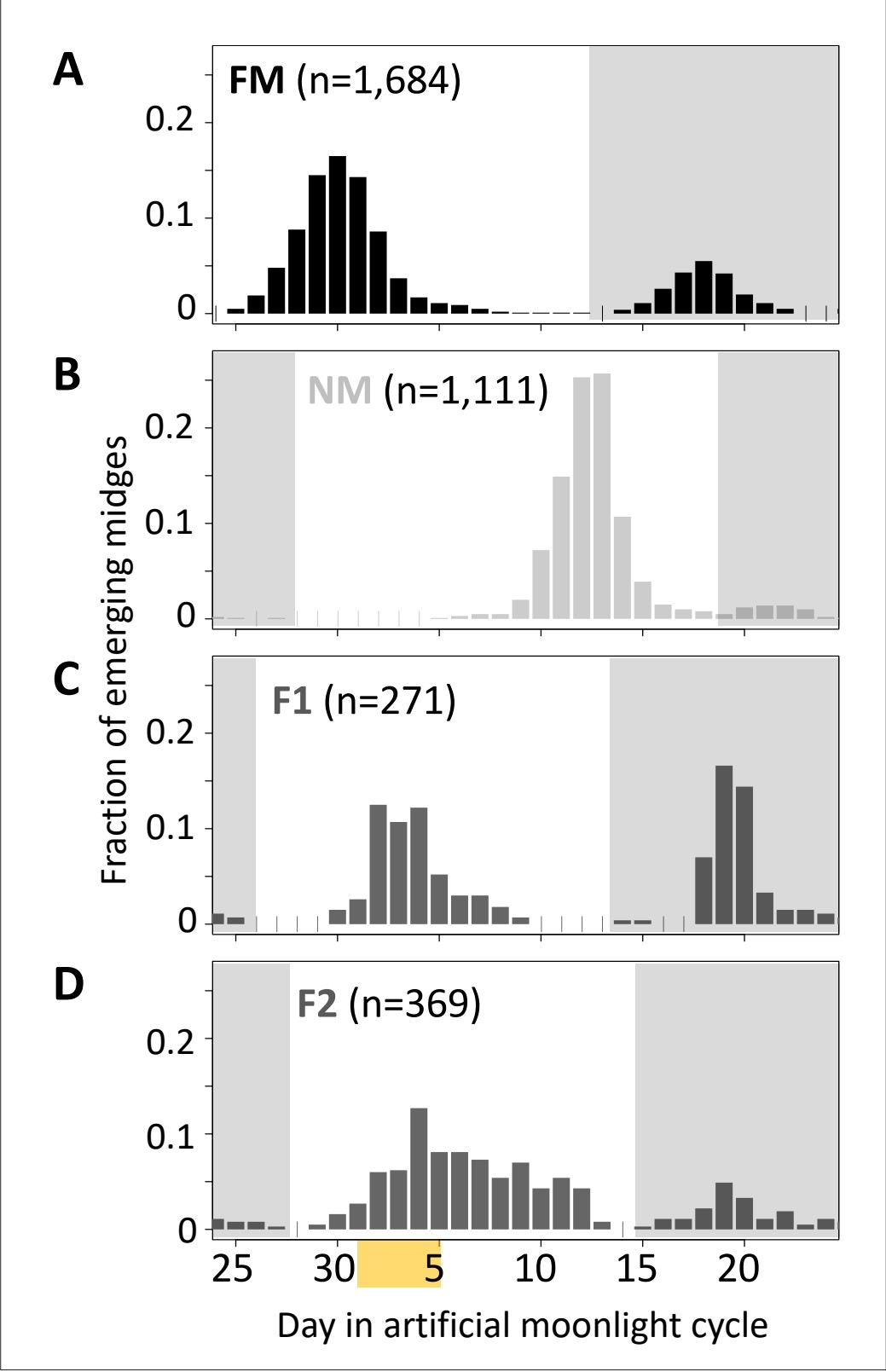

**Figure 4.** Lunar emergence time is heritable. In a cross between FM type (**A**) and NM type (**B**), the F1 hybrids emerges intermediate between the parents (**C**). In the F2 (**D**), the phenotypes spread out again, but do not segregate completely, suggesting that more than one locus controls lunar emergence time. The peaks around day 20 (gray shading) are direct responses to the artificial moonlight treatment (see *Kaiser et al., 2021*). They

*Figure 4 continued on next page*

*Figure 4 continued*
do not occur in the field and were not considered in our analyses. Yellow shading indicates the days with artificial moonlight treatment in the laboratory cultures.

types (see *Figure 1C* and *Kaiser et al., 2021*), we sequenced the full genomes of the two parents in order to identify informative genetic markers for linkage mapping and QTL mapping. We picked a set of 32 microsatellite markers, 23 of which turned out to be reliably amplified and informative in the F2 families, as well as four insertion-deletion polymorphisms (*Supplementary file 4*).

Genetic linkage mapping confirmed the existence of the inversions In(1a) and In(1b), both of which are supported by inverted marker order on the genetic linkage map (*Figure 5A*). Marker order is also inverted for In(2 L), but not for the other inversions (*Figure 5A*). We then performed stepwise QTL identification and Multiple QTL Mapping (MQM) as implemented in R/qtl and found four largely additive QTL for the difference in lunar reproductive timing (*Figure 5A* and *Supplementary file 5*). A specific scan for epistatic effects only shows a weak interaction of chromosome 1 with the QTL on chromosome 2 (*Figure 5—figure supplement 1*). The four QTL model explains 54% of the total variance in lunar reproductive timing and the estimated additive effects of the QTL account for 6 days of the timing difference between the FM and NM type (*Figure 5A*, *Supplementary file 5*). The individual loci explain 6.4%, 10.4%, 20.5%, and 10.8% of the variation in the phenotype (*Supplementary file 5*), but given the size of our mapping family these values could be overestimated by approximately threefold *King and Long, 2017* due to the Beavis effect *Slate, 2013*; *Xu, 2003*. This suggests that the actual variance explained by these loci might be rather in the range of 2–7% and that additional loci of smaller effects are likely to contribute to the phenotype.

In both MQM and additional Composite Interval Mapping (CIM; *Figure 5—figure supplement 2*), we estimated confidence intervals for the detected QTL (*Figure 5A*). There are two QTL on chromosome 1, one of which overlaps with the inversion system. The inversion genotypes of the parents were $SI_2$ and $I_1I_2$ (*Figure 5—figure supplement 3A*), so that in the F1 we can expect partial recombination suppression, depending on the F1 genotypes. This is reflected in reduced map length for the respective genomic region (*Figure 5A*), but there was sufficient recombination to assign the QTL interval to only one end of In(1 a). Based on the first genetic marker outside the QTL's confidence intervals, we matched the QTL region with the corresponding genomic reference sequence (*Figure 5B*). The second QTL on chromosome 1 maps outside the inversion system and is well separated from the other QTL (*Figure 5A and B*). The other two QTL are found on chromosomes 2 and 3 and overlap with In(2 R) and In(3 L), respectively. The parental genotypes for In(2R) are SI and putatively II and for In(3 L) are both SI (*Figure 5—figure supplement 3B and C*), so that we can again expect partial recombination suppression in the F1 and reduced map length. The genomic regions underlying these QTL are large and show several peaks in genetic differentiation (*Figure 5B*). Possibly, the QTL on chromosome 2 and 3 harbor a set of linked loci influencing the lunar emergence phenotype. Taken together, there are at least four unlinked loci which influence this magic trait. Given that the QTL mapping was based on a single pair of parents, the overall genetic architecture in the populations may be even more complex than what we described here.

## Divergent alleles are not associated with specific inversion haplotypes

While some chromosomal inversions show up as genetically differentiated blocks, in all chromosomal inversions there are loci that are far more differentiated than the inversion itself (compare *Figure 2L* and *Figure 3L* to O with *Figure 5B*). This implies that the inversions are not required for protecting genetic differentiation. One explanation for such divergence peaks could be that individual SNPs in the ancestral arrangement are strongly differentiated between the timing types (while it is generally assumed that the inversion is fixed for one allele). Alternatively, there could be recombination and gene conversion placing the divergent alleles also in the inverted haplotypes. In both scenarios individual loci can diverge, while the surrounding inversions remain less differentiated. We tested these scenarios for the most differentiated locus in the genome (*Figure 5B*, red asterisk), which is the *period* locus (*Figure 5C*). We identified an insertion-deletion (indel) mutation in the highly differentiated first intron of the *period* gene (*Figure 5C*, green arrows) and re-genotyped the 48 individuals for this indel (*Figure 5D*; *Figure 5—figure supplement 4*). This confirmed that *period* alleles are indeed strongly differentiated between the FM and NM types. However, we could not find a close association

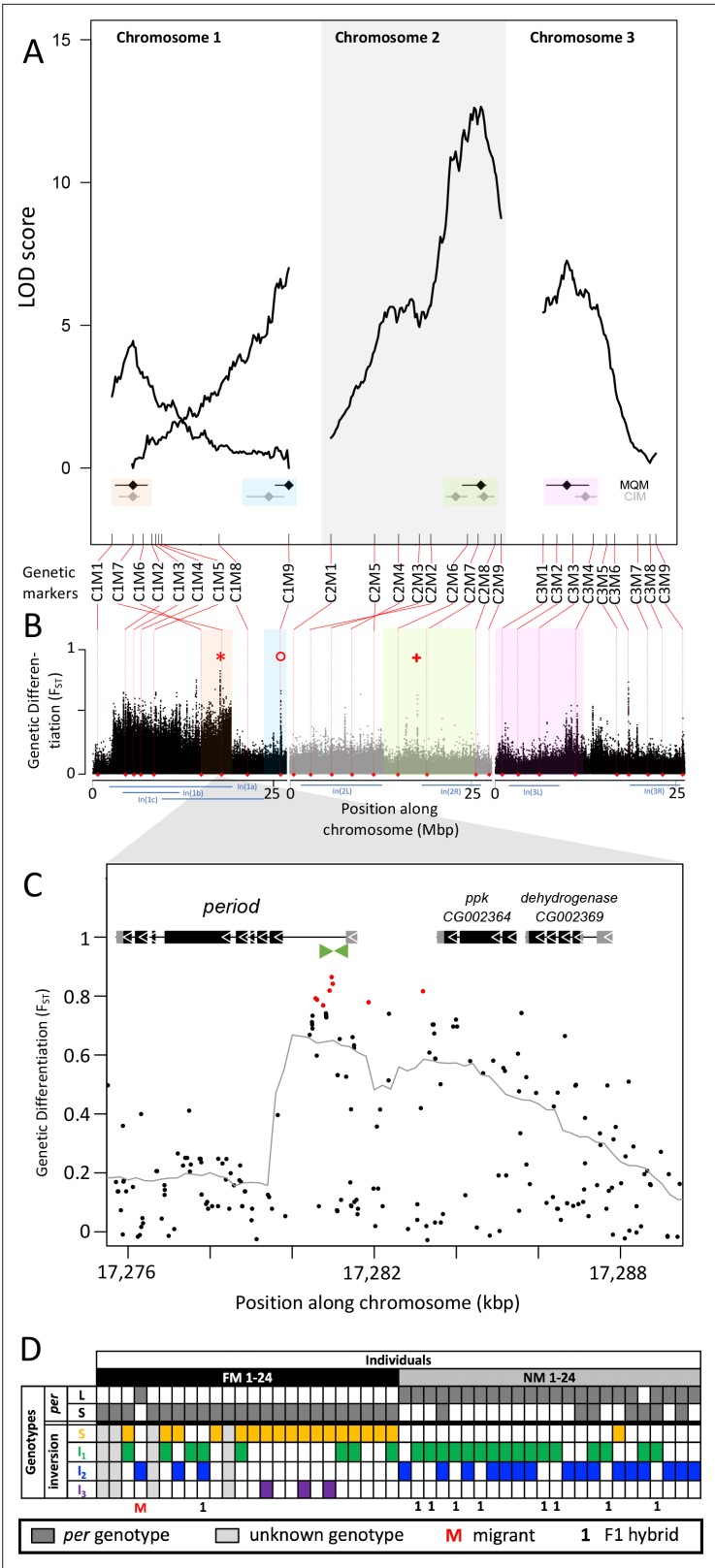

**Figure 5.** Quantitative trait loci (QTL) and genomic regions underlying FM vs NM emergence. (**A**) Multiple QTL Mapping (MQM) identified four significant QTL controlling lunar emergence time. Confidence intervals were determined in MQM and Composite Interval Mapping (CIM). The marker order supports In(1 a) and the nested In(1b) on chromosome 1, as well as In(2 L) on chromosome 2. (**B**) QTL intervals were colour-coded, transferred to

*Figure 5 continued on next page*

*Figure 5 continued*

the reference sequence and assessed for loci which are strongly differentiated ($F_{ST}$) between the FM and NM types. Inversions are represented by blue bars below the plot. The QTL in In(1 a) overlaps with the most differentiated locus in the genome (red asterisk) – the *period* locus. Other markedly differentiated loci within the QTL intervals are the *plum/cask* locus (red circle) and the *stat1* locus (red cross). (**C**) Genetic differentiation is strongest in the first intron and the intergenic region just upstream of the *period* gene. (**D**) The *period* locus was re-genotyped based on an insertion-deletion mutation in the first intron (green arrows in (**C**)). The long (**L**) and short (**S**) *period* alleles are only loosely associated with inversion haplotypes.

The online version of this article includes the following source data and figure supplement(s) for figure 5:

**Figure supplement 1.** Scan for interacting QTL.

**Figure supplement 1—source data 1.** Table of test statistics for epistatic interactions.

**Figure supplement 1—source data 2.** Table of test statistics for additive interactions.

**Figure supplement 2.** LOD profiles from Composite Interval Mapping (CIM).

**Figure supplement 3.** Inversion genotypes of the parents of the FMxNM cross.

**Figure supplement 4.** Genotyping of an insertion-deletion in the *period* locus.

**Figure supplement 4—source data 1.** Original gel pictures of the genotyping of an insertion-deletion in the *period* locus.

**Figure supplement 5.** Gene models and genetic differentiation at the *plum* locus.

**Figure supplement 6.** Gene models and genetic differentiation at the *stat1* locus.

between the *period* alleles and the inversion haplotypes of chromosome 1 (*Figure 5D*). Both alleles occur is several inversion haplotypes, confirming that there is gene flux between standard and inversion haplotypes. This implies that differentiation of the *period* locus is not driven by differentiation of the inversion system, that is it is not enhanced by LD with other locally adaptive variants.

## Candidate loci

The overlap of genetic divergence peaks and the QTL's 95% Bayesian intervals, allows us to identify potential candidate genes for controlling the magic trait, i.e. phenotypic divergence in lunar reproductive timing (see *Supplementary file 6* for an overview). Some loci are particularly conspicuous in that they are by far the most divergent within a specific QTL. The QTL in the inversion system on chromosome 1 is narrow and harbors a single strongly divergent locus, which is the *period* locus (*Figure 5B*, red asterisk). The *period* locus is the most differentiated locus in the entire genome (maximum $F_{ST} = 0.86$) and *period* is a core circadian clock gene. The QTL at the end of chromosome 1, outside the inversion system, also shows a single strong divergence peak (*Figure 5B*, red circle). This is the first intron of the *plum* gene (*Figure 5—figure supplement 5*). The large intron likely contains regulatory regions for both *plum* and the directly adjacent *cask* gene. On chromosome arm 2 R the QTL has several divergence peaks, but the strongest peak hits the *stat1* gene (*Figure 5B*, red plus sign; *Figure 5—figure supplement 6*). *Plum, cask,* and *stat1* are all involved in nervous system development in *Drosophila melanogaster Yu et al., 2013*; *Gillespie and Hodge, 2013*; *Ngo et al., 2010*, suggesting that divergence in lunar reproductive timing may involve nervous system remodeling as well as circadian timekeeping. These gene functions are plausible to be involved in lunar time-keeping and congruent with the results from a companion study *Fuhrmann et al., 2023*, which assessed the loss of lunar timing in a different set of *Clunio* populations. Thus, evidence from gene function, genetic divergence and QTL mapping align to support these candidate genes.

## Discussion

We found multiple large and polymorphic chromosomal inversions in the sympatric FM and NM strains, covering all chromosome arms of the *C. marinus* genome. This is consistent with large non-pairing and inverted regions observed in *C. marinus* polytene chromosomes *Michailova, 1980*. These inversions reduce recombination, as is observed in large LD blocks (*Figure 2—figure supplement 1* and *Figure 3—figure supplement 1*), and possibly lock several loci involved in divergent reproductive timing into supergenes (*Figures 2A and 3A*). However, most of these inversions are not differentiated between timing types and in all inversions, there are loci more divergent than the inversion itself. We

show that divergent alleles at the *period* locus are not associated with specific inversion haplotypes or the standard haplotype (*Figure 5D*). Thus, there must be considerable recombination and/or gene conversion inside the inversion, placing the *period* alleles in different inversion haplotypes. This is likely also true for other divergent loci and forms the basis for genetic differentiation at individual loci to exceed the inversion background. We therefore propose that in this study system genetic linkage is not the major mechanism for coupling ecological divergence and assortative mating. Instead, we favor the magic trait scenario, in which recombination does not matter, because the relevant components of reproductive isolation are influenced by a single trait. Given a magic trait, all loci influencing the trait will be under divergent natural selection. We thus assume that peaks of genetic differentiation include the loci responsible for ecological adaptation, and that their differentiation is driven by permanent divergent natural selection. As a consequence, physical linkage between adaptive alleles and inversion haplotypes, though incomplete, may drive the spread of specific inversion haplotypes and thus differentiation of the inversion system on chromosome 1. Differentiation of the inversion system then creates a genetic substrate for further components of reproductive isolation to evolve and be genetically linked to the existing loci responsible for reproductive isolation. Eventually, this may lead to the completion of speciation.

At the same time, QTL mapping revealed at least four independent loci underlying lunar reproductive timing. Therefore, we are neither dealing with a single pleiotropic locus that affects several components of reproductive isolation, nor with a single supergene that would combine genetic variants underlying different components of reproductive isolation, such as local adaptation and assortative mating. Our data show that an oligogenic architecture can underlie ecological and reproductive divergence in sympatric populations. In addition, the four QTL we identified here for the difference in lunar reproductive timing between the FM and NM types in Roscoff do not overlap with the two QTL we identified previously for difference in lunar reproductive timing between Normandy (Por-1SL) and Basque Coast (Jean-2NM) strains *Kaiser et al., 2016*; *Kaiser and Heckel, 2012*. This indicates that within the species *C. marinus*, different populations have found different genetic solutions for adapting lunar reproductive timing to the local tidal regime. Given that both studies relied on a single crossing family, the overall genetic basis of lunar reproductive timing may even be more complex than what we have shown here.

During the last ice age, the English Channel did not exist *Patton et al., 2017*, setting the time frame of establishment of the sympatric Roscoff populations roughly to the last 10,000 years. Genetic differentiation between the inversion haplotypes (*Figure 2—figure supplement 3*) is higher than what would be expected to evolve in this time-frame. We thus assume that the inversions were already segregating in the ancestral populations or may have recently introgressed from a different geographic site. The latter might also be true for the genetic variants underlying sympatric ecological divergence.

The detection of *period* as the most differentiated locus between NM and FM types, overlapping with a QTL for lunar reproductive timing, suggests that this circadian clock gene might also be involved in lunar time-keeping. The other prominent candidates, *plum*, *cask,* and *stat1* are involved in neuronal or synaptic development and plasticity. Notably, *cask* is a direct interaction partner of CaMKII *Gillespie and Hodge, 2013*, which we previously found to be involved in circadian timing differences in *Clunio Kaiser et al., 2016*. Hence, both QTL on chromosome 1 suggest a close connection of circadian and circalunar time-keeping. This is backed by recent findings in Baltic and Arctic *Clunio* strains, for which genomic analysis of the loss of lunar rhythms also implied circadian GO terms as the top candidates for affecting circalunar time-keeping, followed by GO terms involved in nervous system development *Fuhrmann et al., 2023*. Thus, our data not only elucidate the genetic architecture underlying a magic trait, but also give hints to the genetic basis of the yet enigmatic circalunar clock.

# Methods
## Sampling and laboratory strains
Laboratory strains for crossing experiments and field samples for whole genome sequencing of 48 individuals were available from a previous study *Kaiser et al., 2021* (FM type = Ros-2FM; NM type = Ros-2NM). Laboratory strains were kept in standard culture conditions *Neumann, 1966* at 20 °C and

under a light-dark cycle of 16:8. An artificial moonlight cycle with four nights of dim light every 30 days served to synchronize reproduction in the laboratory strains.

## Sequencing, read mapping, and genotype calling

DNA from the 48 field caught individuals were subject to whole genome sequencing in the Max Planck Sequencing Centre (Cologne) on an Illumina HiSeq3000 according to standard protocols. Independent sequencing runs were merged with the *cat* function. Adapters were trimmed with Trimmomatic *Bolger et al., 2014* using the following parameters: ILLUMINACLIP <Adapter file>:2:30:10:8:true, LEADING:20, TRAILING:20, MINLEN:75. Overlapping read pairs were merged with PEAR *Zhang et al., 2014* using -n 75 c 20 k and mapped to the Cluma_2.0 reference genome (available in the Open Research Data Repository of the Max Planck Society under DOI 10.17617/3.42NMN2; manuscript in preparation) with bwa mem *Li and Durbin, 2009* version 0.7.15-r1140. Mapped reads were merged into a single file, filtered for -q 20 and sorted with samtools v1.9 *Li et al., 2009*. SNPs and small indels were called using GATK v3.7–0-gcfedb67 *McKenna et al., 2010*. All reads in the q20 sorted file were assigned to a single new read-group with 'AddOrReplaceReadGroups' script with LB = whatever PL = illumina PU = whatever parameters. Genotype calling was then performed with HaplotypeCaller and parameters `--emitRefConfidence` GVCF -stand_call_conf 30, recalibration of base qualities using GATK BaseRecalibrator with '-knownSites'. Preparing recalibrated BAM files with GATK PrintReads using -BQSR. Recalling of genotypes using GATK HaplotypeCaller with previously mentioned parameters. Individual VCF files were combined into a single file using GATK GenotypeGVCFs.

## Genetic differentiation (F$_{ST}$)

The vcf file containing GATK-called SNPs and small indels from 48 Ros-2FM and Ros-2NM males was filtered with vcftools version 0.1.14 *Danecek et al., 2011* for minor allele frequency of 0.05, maximum of two alleles, minimum quality (minQ) of 20, maximum missing genotypes of 20%, which finally left 721,000 variants. Genetic differentiation between the two populations (F$_{ST}$) was estimated using vcftools parameters `--weir-fst-pop --fst-window-size 1 --fst-window-step 1`.

## Long-range linkage disequilibrium (LD)

Linkage disequilibrium was calculated between all variants along the three chromosomes in each of the populations in search of the signatures of large genomic inversions. Vcf file containing GATK-called SNPs and small indels from 24 Ros-2FM males was filtered for minor allele frequency of 0.20, maximum of two alleles, minimum quality (minQ) of 20, maximum missing genotypes per site of 20% leaving 344,331 variants. The same was done for the 24 Ros-2NM individuals resulting in 352,915 variants. The filtered vcf files were converted to plink input files and plink version 1.90 beta *Chang et al., 2015* was used to calculate linkage disequilibrium between all variants with parameters `--r2 --inter-chr`. The large LD file was then sorted into 3 files, one for each chromosome. Breakpoints of large chromosomal inversions were approximated by plotting r$^2$ and manually looking for obvious breaks in the r$^2$ scores (*Figure 2—figure supplement 1* and *Figure 3—figure supplement 1*; *Supplementary file 2*).

## Principal component analysis (PCA)

PCA was run for the entire genome, for sliding windows along the chromosome ('windowed PCA'), and for windows corresponding to the inversions ('local PCA'). The vcf file containing GATK-called SNPs and small indels from 48 Ros-2FM and Ros-2NM males was filtered with vcftools version 0.1.14 *Danecek et al., 2011* for minor allele frequency of 0.05, minimum quality (minQ) of 20, maximum missing genotypes per site of 20%, leaving 703.579 variants. Principal component analysis was calculated with plink version 1.90 beta *Chang et al., 2015* using flags `--nonfounders --pca var-wts --chr-set 3 no-xy no-mt`. Windowed PCA: In order to investigate regions of the genome with unusual degree of variance, PCA was run in windows along the three chromosomes. Vcf files were subdivided into vcf files containing variants that belong to 500 kb sliding windows with 100 kb steps. They were further used to calculate PCA values for each window as noted above. Local PCA: To get the genotype-estimate of the identified inversions (see long-range disequilibrium section), we subdivided vcf files according to the estimated inversion breakpoints (*Supplementary file 2*), and

calculated principal components 1 and 2 as described above. Number of variants that belong to each window are listed in *Supplementary file 3*.

## Admixture

Ancestry and relatedness of the 48 individuals of Ros-2FM and Ros-2NM population were investigated with admixture version 1.3.0 *Alexander et al., 2009*. Admixture was run for the entire genome, for sliding windows along the chromosome ('windowed admixture'), and for windows corresponding to the inversions ('local admixture'). We used vcf files previously described in the PCA section as input. Admixture was calculated using k of 2–4. Windowed admixture: In order to identify regions of the genome with ancestry different from the general population, we ran admixture on vcf files containing variants that belong to 500 kb sliding windows with 100 kb steps. Local admixture: In order to genotype the inversions (see long-range disequilibrium section) we calculated admixture using vcf files containing only variants from those regions of the genome (*Supplementary file 2*; *Supplementary file 3*).

## Observed heterozygosity

In order to complement the windowed and local PCA and admixture analyses and further corroborate the indirect genotyping of the chromosomal inversions, we calculated observed heterozygosity using vcftools version 0.1.14 *Danecek et al., 2011* with `--het` flag. Proportion of observed heterozygosity was calculated for each file by dividing the number of observed heterozygotes with the number of variants. Patterns of heterozygosity along the chromosomes were assessed based on the vcf files containing variants that belong to 500 kb sliding windows with 100 kb steps (see PCA and admixture section). Finally, in order to calculate observed heterozygosity of the chromosomal inversions, observed heterozygosity was calculated per inversion window for each individual (*Supplementary file 2*; *Supplementary file 3*). Then observed heterozygosity was calculated and plotted for each inversion genotype according to local PCA and admixture.

## Crosses, phenotyping, and genotyping

After synchronizing the NM and FM strains by applying different moonlight regimes, single pair crosses were performed. F1 egg clutches were reared individually and the emerging adults were allowed to mate freely within each F1 family, leading to several sets of F2 families that go back to a single pair of parents. One of these sets was picked for QTL mapping. As the second peak is not under clock control, but a direct response to moonlight *Kaiser et al., 2021*, analysis was restricted to individuals emerging in the first peak (n=158; compare non-shaded area in *Figure 4D*.). In the F2, the peaks are clearly separated from each other by days without any emergence. DNA was extracted with a salting out method *Reineke et al., 1998* and amplified with the *REPLI-g Mini Kit* (Qiagen) according to the manufacturer's instructions. The two parents were subject to whole genome sequencing with 2x150 bp reads on an Illumina HiSeq2000 according to standard protocols. Read mapping and genotype calling were performed as described above. Microsatellite and indel genotypes were obtained by custom parsing of the vcf files. PCR primers (*Supplementary file 4*) for amplifying the microsatellite and indel regions were designed with Primer3. Indels were PCR amplified and then run and scored on 1.5% agarose gels. Microsatellites were amplified with HEX- and FAM-labeled primers and run on an ABI PRISM 3100 Genetic Analyzer. Chromatograms were analyzed and scored with in R with the *Fragman* package *Covarrubias-Pazaran et al., 2016*. The resulting genotype matrix can be found in the R/qtl input file (*Supplementary file 7*).

## Linkage and QTL mapping

Linkage analysis and QTL mapping were based on a set of F2 families derived from a single pair of parents. Analyses were performed in R/qtl according to the script in *Source code 1*. Briefly, the recombination fraction (*est.rf*) and distribution of alleles (*checkAlleles*) were assessed to confirm the quality of the input data. The correct marker order was inferred by rippling across all markers of each chromosome (*ripple*). Missing data (*countX0*) and error load (*top.errorload*) were assessed. Then the best model was obtained in a stepwise selection procedure (*stepwiseqtl*). Additional interactions were checked for in a two QTL scan (*scantwo*), but were negligible and not considered in the final model. Finally, a model with 4 non-epistatic QTL was subject to model fitting for Multiple QTL Mapping

(*fitqtl*) and the QTL's 95% Bayesian confidence intervals were estimated (*bayesint*). Composite Interval Mapping was performed in QTLcartographer *Bioinformatics Research Center, 2006* with various selection procedures (forward, backward), exclusion windows (10 cM and 20 cM) and covariates (3, 5, 10; *Figure 5—figure supplement 2*). The LOD significance threshold was estimated by running 1000 permutations and a p value of 0.05. For the QTL locations in CIM the 1 LOD intervals were plotted.

## PacBio data and structural variant calling

PacBio long read data was obtained for pools of 300–500 individuals from laboratory strains of Ros-2NM and Ros-2FM (Roscoff, France). DNA was extracted as above and sequenced with standard protocols on a PacBio Sequel II at the Max Planck Sequencing Facility in Cologne, Germany. Raw long-reads of Ros-2FM and Ros-2NM were mapped against the reference CLUMA 2.0 (publication in preparation) using NGM-LR v.0.2.7 *Sedlazeck et al., 2018* with default settings. Alignments were sorted, filtered (q20) and indexed with samtools v.1.9 *Li et al., 2009*. Three different SV-calling tools were used per population to discover SVs. Sniffles v.1.0.11 *Sedlazeck et al., 2018* was run with parameters `-- min_het_af 0.1` and `–genotype`. SVIM v1.2.0 *Heller and Vingron, 2019* was run with default parameters (svim alignment). Finally, Delly v0.8.6 *Rausch et al., 2012* was run with parameters `lr -y pb -q 20 --svtype` (INS, insertions; INV, inversions; DUP, duplications; DEL, deletions) and the output VCF files per SV type were merged with a custom script. The resulting VCF files were then sorted using vcf-sort and filtered for quality "PASS" using a custom bash script. SURVIVOR v1.0.6 *Jeffares et al., 2017* was used to filter variants for a minimum size of 300 bp and at least 5 reads supporting each variant (SURVIVOR filter NA 300–1 0 5). BND variants detected with Sniffles and SVIM calls were excluded with a custom bash script before using SURVIVOR merge (options set to 50 1 1 1 0 0 300) on all VCFs produced per population. The merged VCFs were then used as an input to reiterate SV calling with Sniffles using the same parameters as above plus `--Ivcf` option and `--min_support 5` (minimum number of reads supporting a SV). Finally, SURVIVOR merge (same parameters as above) was used to merge the re-genotyped SVs detected in Ros-2NM and Ros-2FM. SV support was reported in *Figures 1 and 2* if there was a breakpoint detected within 3 kb of the boundaries of the LD blocks. The full set of SV calls is given in *Supplementary file 1*.

## *Period* genotyping

An insertion-deletion (indel) mutation in the *period* locus was genotyped. The fragment was PCR amplified (primers: 5'-GAATACTGAGTGTAAGACTTGGC and 5'-ACAACGTGACCTGTGACAAT) and run and scored on 1.5% agarose gels.

## Materials and correspondence

Requests should be addressed to Tobias S. Kaiser (kaiser@evolbio.mpg.de).

## Acknowledgements

Tjorben Nawroth, Kerstin Schäfer, Susanne Mentz, Elke Bustorf and Sina Schirmer provided laboratory assistance. Jürgen Reunert provided animal care. We thank all members of the research group *Biological Clocks*, as well as Diethard Tautz for feedback and discussion. This work was supported by the Max Planck Society via an independent Max Planck Research Group and by an ERC Starting Grant (No 802923) awarded to TSK. CP was funded by the International Max Planck Research School (IMPRS) for Evolutionary Biology.

## Additional information

### Funding

| Funder | Grant reference number | Author |
| --- | --- | --- |
| European Research Council | 802923 | Tobias S Kaiser |
| Max-Planck-Gesellschaft | | Tobias S Kaiser |

| Funder | Grant reference number | Author |
|---|---|---|

The funders had no role in study design, data collection and interpretation, or the decision to submit the work for publication. Open access funding provided by Max Planck Society.

### Author contributions
Dušica Briševac, Carolina M Peralta, Formal analysis, Writing – review and editing; Tobias S Kaiser, Conceptualization, Formal analysis, Supervision, Funding acquisition, Visualization, Methodology, Writing – original draft, Project administration

### Author ORCIDs
Carolina M Peralta ⬮ http://orcid.org/0000-0002-8058-4952
Tobias S Kaiser ⬮ http://orcid.org/0000-0002-4126-0533

### Decision letter and Author response
Decision letter https://doi.org/10.7554/eLife.82825.sa1
Author response https://doi.org/10.7554/eLife.82825.sa2

## Additional files

### Supplementary files
• Supplementary file 1. Genome-wide structural variant (SV) calls.
• Supplementary file 2. Inversion breakpoints as estimated from long range LD.
• Supplementary file 3. Number of variants in inversion windows.
• Supplementary file 4. Microsatellite and length polymorphism markers.
• Supplementary file 5. Final QTL model as given by the *fitqtl* function, obtained with multiple imputation, a normal phenotype model and based on 158 observations.
• Supplementary file 6. Peaks in genetic differentiation ($F_{ST}$).
• Supplementary file 7. r/QTL input file.
• MDAR checklist
• Source code 1. r/QTL script for QTL mapping.

### Data availability
Sequencing data was submitted to ENA under project number PRJEB54033. The CLUMA2.0 reference genome is available on the Open Research Data Repository of the Max Planck Society (EDMOND) under https://doi.org/10.17617/3.42NMN2.

The following dataset was generated:

| Author(s) | Year | Dataset title | Dataset URL | Database and Identifier |
|---|---|---|---|---|
| Peralta CM, Kaiser TS | 2023 | Roscoff FM and NM individuals | https://www.ebi.ac.uk/ena/browser/view/PRJEB54033 | EBI European Nucleotide Archive, PRJEB54033 |

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
