## [Editor Report]

This important article provides solid evidence for an apparent oligogenic architecture of an ecologically relevant trait, the circalunar reproduction of marine midges, which contributes to assortative mating, is likely under divergent selection, and supports reproductive isolation in sympatry.

---

## [Decision Letter]

**Decision letter after peer review:**

Thank you for submitting your article "An oligogenic architecture underlying ecological and reproductive divergence in sympatric populations" for consideration by *eLife*. Your article has been reviewed by 2 peer reviewers, and the evaluation has been overseen by a Reviewing Editor and Detlef Weigel as the Senior Editor. The reviewers have opted to remain anonymous.

The reviewers have discussed their reviews with one another, and the Reviewing Editor has drafted this to help you prepare a revised submission. As you will see, both expert reviewers like the topic and the system you have established, but have also significant concerns, mostly about interpretation, but some also regarding technical issues.

Essential revisions:

Please be more stringent in your discussion and interpretation of 'magic traits' and their genetic basis, and especially how they can be differentiated from related genetic and evolutionary phenomena.

State clearly that the conclusions are drawn from a single cross rather than population-wide linkage mapping and that it cannot be excluded that the species-wide genetic architecture of the trait of interest is more complex than suggested here.

Temper your claims of genomic regions of high differentiation being the causal drivers of the ecological differentiation.

Add technical detail, as outlined in the individual reviews.

*Reviewer #1 (Recommendations for the authors):*

This manuscript describes a very interesting set of experiments on co-located populations of marine midge that emerge either at full or at new moon. There are several aspects of the study that will attract broad attention: the genetic architecture of a trait involved in reproductive isolation, the role of chromosomal inversions in population divergence and the link between the circadian and circalunar clocks. The authors choose to focus on 'magic traits'. This is not a straightforward idea. For a broad audience, it is important that the idea is explained carefully and I do not think that the current Introduction provides such a clearly-argued context. I have made specific comments by line numbers below. A related issue is whether the timing difference is actually under divergent selection. Probably it is but this is important for the argument in the manuscript and so reasons for expecting divergent selection and the likely nature of the selection should be described.

The experiments and analyses generally appear robust and appropriately interpreted. I have some reservations that are detailed below but I doubt any of them will influence the major conclusions. Data are not presented for long-read detection of SVs. There is generally some uncertainty about the interpretation of inversions, but this is handled well. The inversion genotypes of individuals used in the QTL mapping crosses were not determined (or at least they are not presented). This influences the direct interpretation of the crosses but it may also influence the larger picture because the mapping cross is just between two individuals. If these individuals were homozygous for the same arrangement of a given inversion, the cross might give a very different outcome from one between individuals homozygous for different arrangements. This is a particular case of the general problem of interpreting a cross between two individuals as if it represents the genetic basis of the difference between two populations.

More specific comments follow by line number:

30-32 – this is not a great summary because it starts by contrasting two extremes and then effectively says that there is evidence for some kind of intermediate. It is surprising that Coyne and Orr's book is not cited since it is probably the best summary of the debate up to 2004.

33 – 'mechanisms' is not a good choice of word here since it implies something that is fabricated by selection for the purpose of isolation

34-36 – also cite Felsenstein 1981. Note that 'ecological divergence' is a component of 'reproductive isolation' and so the wording here is problematic. It should be something like 'ecological divergence and other components of reproductive isolation'. This applies at multiple points in the manuscript.

37 – while ref. 8 is the origin of the term 'magic trait', ref. 13 provided an important discussion of the idea and ref. 6 provided an alternative terminology as well as clarifying the relationship with pleiotropy. If the magic trait/multiple-effect trait idea is central to this manuscript, some of this literature should be cited.

41 – see ref. 6 for a view on the role of pleiotropy. In my view, nothing in the magic trait idea implies a small number of genes: the point is that the trait contributes to multiple components of reproductive isolation and this can be true regardless of its genetic basis. This is part of the reason why confounding the idea with pleiotropy is unhelpful.

42-44 – the theoretical basis for this claim should be supported with citations, as well as giving examples in the next sentence.

65 – actually, this is not enough to meet the magic trait definition because there also need to be evidence for 'ecological divergence': the FM and NM types need to be locally adapted to their niches in some sense. Alternatively, hybrids might suffer a fitness cost – a form of incompatibility. This would fit the multiple-effect trait definition advanced by ref. 6 but not the definition of magic traits used here (although I think it might fit the definition originally used by Gavrilets).

72 – I think the authors probably mean 'linkage disequilibrium' here, rather than 'genetic linkage'.

91-2 – some caution is needed here since F1 and BX hybrids alone do not represent gene flow. A plot of heterozygosity against hybrid index, or an analysis using NewHybrids would help to show whether later generation hybrids occur. However, caution is also needed because of segregation of structural variants that might be shared between ecotypes and can distort Fst and admixture analyses.

98-100 – meaning 'polymorphic in both timing types', I assume. It seems that the data behind the SV calls in figure 1C are not shown anywhere.

Figure 1 and S4 – running the windowed PCA and the PCAs in Figure 1 without the 4 outlying FM individuals might well give clearer patterns. The rationale behind the grouping into genotypes in Figure 1 is not clear.

134-140 – support for I3 is relatively weak, but that is clear from the text.

141-6 – this partly answers the question about assignment to genotypes – it uses more than just the PCA shown in Figure 1. At present, this is all quite hard to access, even though I am used to plots of this type.

150-2 – it may be worth noting that this helps to explain the LD patterns in Figure S2: because NM segregates only for I1 and I2, there is LD only in the I2 region (the only region where recombination is suppressed). In FM, recombination is suppressed over the length of I1, but we might expect more gene flux in the area of I2 because it is collinear in SI2 individuals. However, this is all without thinking about gene flow between FM and NM. The strong differences in LD seem quite surprising to me, given the previous inference of extensive gene flow.

Table S1 gives the margins of the area of SD (with surprisingly high precision). Since LD can extend well beyond breakpoints it may be better not to think of these as estimated breakpoint positions.

158 – again, SV calling data are not shown.

Figure 3 – it is unfortunate that the 'peaks around day 20' overlap with the 'natural' distributions, and no single cut-off can separate them. Do points around day 25 belong to the 'spurious' distribution? Are conclusions about QTL sensitive to decisions about boundaries?

168 – maybe 'at least four QTL' since it is impossible to exclude the presence of other loci or of more loci under the broad peaks observed (indeed, Figure S8 does suggest multiple loci under some peaks).

Figure 4A,B – it is not clear why two lines are plotted in 4A for Chr1. The inference that the gene order is 'reversed' in I1 and I2 is relative to the reference genome (and actually quite hard to see in the current plot). Importantly, there is no evidence of recombination suppression in the cross analysed, which implies that the FM and NM parents were homozygous for the same (inverted) arrangement. This is actually quite surprising, given the arrangement frequencies inferred from Figure 1. However, it is unclear whether the methods used would have picked up partial recombination suppression due to heterozygosity in the grandparents (leading to heterozygosity in some but not all of the F1 individuals).

185 – does 'largely additive' mean that dominance effects were small or not significant? In Table S4d, the dominance effects look non-significant but there is no comparison of models with or without dominance. Fig, S7B,C have not come through well in my PDF. Here the tests for epistasis seems to exclude dominance.

187-90 – this, of course, confirms that other loci are likely to contribute.

193-5 – see comments above on Figure 4A,B. It needs to be clear what the inferred inversion genotypes of the parents and F1 were in these crosses. This influence interpretation substantially.

199 – but these inversions were also apparently not segregating in the cross families.

207 – Figure 4B does not make this clear because it does not show Fst for each inversion, only for SNPs within inversion. Also note that a new inversion is typically fixed for one allele at any locus where the ancestral arrangement is polymorphic. There are then many combinations of inversion and SNP frequencies in two populations for which Fst is higher for the SNP than for the inversion, without any requirement for gene flux (as suggested on line 209). Gene flux is needed for cases where both arrangements carry both alleles (as seems to happen in some cases in Figure 4D, although the situation here is complicated by the putatively overlapping inversions which can enhance gene flux).

217-8 – there is no expectation that differentiation at a single locus will be protected by an inversion so it would help to reword this sentence a bit. I think the authors mean that differentiation at period is not enhanced by association with other locally-adaptive variants that is maintained by an inversion.

250-1 – this comes back to issues in the Introduction with the 'magic trait' idea. The basis of this idea is that recombination is less important because recombination cannot separate the two (or more) components of reproductive isolation influenced by the trait. If one component of isolation is local adaptation (as proposed here but not directly established) then each locus influencing the trait will be under divergent selection. An inversion might be favoured because it maintains advantageous combinations of these loci (the Kirkpatrick-Barton mechanism) but this is different from the case where assortative mating (for example) is not under direct selection and so only evolves in response to indirect selection (as in classical models of reinforcement).

251-3 – this is a slightly odd distinction. I think it needs to be explained a bit more fully and, ideally, related to relevant models for the spread of inversions.

265-7 – this comment about what is 'generally assumed' should be supported by citations. It reflects a view in the Introduction that 'magic traits' are expected to have a simple genetic basis, a view that I consider incorrect (see earlier comments).

272-3 – I am not sure that it is helpful to introduce the idea of 'evolutionary branching' here without further discussion. It would be more in line with the rest of the manuscript to say that the timing ecotypes might accumulate other components of reproductive isolation and eventually complete speciation.

287-8 – in line with previous comments, I think that describing the genetic basis of a putative ‘magic trait’ is a valuable contribution but does not require any re-evaluation.

Methods

328-30 – this identifies the margins of the region of LD, which do not necessarily correspond to breakpoints.

372-3 – this does not provide sufficient information – see comments on the Results.

384-5 – this analysis used all F2 families derived from one pair of grandparents, I believe. This should be made explicit. The inversion genotypes of the parents should be determined (and ideally of the F1 individuals, but this would have to be inferred from their offspring, I think), because the parents may have been heterozygous and so the F1 may have varied.

389-90 – so far as I can see this was without formal testing for dominance effects.

395-413 – it seems that the results of these analyses are not available anywhere.

*Reviewer #2 (Recommendations for the authors):*

This study system promises to hold exciting insights into the genetic basis of ecological traits but there are limitations in the analyses and interpretation that temper my overall enthusiasm for the paper. These are included in my public review but I list them with specific analysis suggestions below:

1) The QTL mapping relies on a modest number of individuals and there are important details missing. The authors should provide the overall heritability of the trait, which can be estimated from their parental and offspring data with some assumptions. For the estimates of QTL effect sizes from such small samples, there is a common issue known as the Beavis effect that can inflate the effect size of individual QTL. Related to this, the total effect size of the QTL reported is extremely high and the authors should consider (e.g. with simulations) that the effect size is substantially inflated. The authors should report individual effect sizes and explore their power to detect QTL with different effect sizes given their study design. I think that consideration of heritability and overall power will help with interpretation of this result.

2) My major issue with the paper is the interpretation of divergence within the inversion as linked causally to the genes underlying ecological divergence. As the authors observe, divergence will vary within an inverted region. This can be traced to myriad factors, including variation in mutation rate, variation in constraint, patterns of ancestral polymorphism within this region, and variation in gene conversion within the inversion. Given this, I do not think it is valid to interpret the regions of high differentiation as the causal drivers of the ecological differentiation. In my view, the results need to be reframed in the context of ecological adaptation dependent on large inversions and potentially interesting candidate genes, without a claim that the results can be narrowed to individual genes.

3) The authors imply in the discussion based on historical results that the ecotype evolved in situ in the last ~60 years. This speculation should be removed as it seems much more likely that the ecotype was either missed in sampling or a recent migrant introduced it to the area in the recent past.

---

## [Author Response]

Essential revisions:Please be more stringent in your discussion and interpretation of 'magic traits' and their genetic basis, and especially how they can be differentiated from related genetic and evolutionary phenomena.

We have completely rewritten the introduction as well as the discussion, in order to now make a clear distinction between magic traits, pleiotropy and genetic linkage as mechanisms for coupling different components of reproductive isolation.

State clearly that the conclusions are drawn from a single cross rather than population-wide linkage mapping and that it cannot be excluded that the species-wide genetic architecture of the trait of interest is more complex than suggested here.

We have included respective statements in the results (lines 209-218; 259-261), discussion (lines 386-398) and method sections (lines 481-482).

Temper your claims of genomic regions of high differentiation being the causal drivers of the ecological differentiation.

We tempered our wording and now speak of “potential candidate genes” which “may” point to specific molecular mechanisms to be involved in ecological divergence.

Add technical detail, as outlined in the individual reviews.

We have added technical detail in many sections, as requested by the reviewers (for details see specific replies below).

Reviewer #1 (Recommendations for the authors):This manuscript describes a very interesting set of experiments on co-located populations of marine midge that emerge either at full or at new moon. There are several aspects of the study that will attract broad attention: the genetic architecture of a trait involved in reproductive isolation, the role of chromosomal inversions in population divergence and the link between the circadian and circalunar clocks.

We thank the reviewer for this positive assessment of our work.

The authors choose to focus on 'magic traits'. This is not a straightforward idea. For a broad audience, it is important that the idea is explained carefully and I do not think that the current Introduction provides such a clearly-argued context. I have made specific comments by line numbers below.

We have entirely revised the introduction in order to provide a clearer context (lines 30-62), based on the very helpful detailed comments that the reviewer gave below.

A related issue is whether the timing difference is actually under divergent selection. Probably it is but this is important for the argument in the manuscript and so reasons for expecting divergent selection and the likely nature of the selection should be described.

We have inserted the reasoning for why divergent selection is expected (lines 75-88).

The experiments and analyses generally appear robust and appropriately interpreted. I have some reservations that are detailed below but I doubt any of them will influence the major conclusions. Data are not presented for long-read detection of SVs. There is generally some uncertainty about the interpretation of inversions, but this is handled well. The inversion genotypes of individuals used in the QTL mapping crosses were not determined (or at least they are not presented). This influences the direct interpretation of the crosses but it may also influence the larger picture because the mapping cross is just between two individuals. If these individuals were homozygous for the same arrangement of a given inversion, the cross might give a very different outcome from one between individuals homozygous for different arrangements. This is a particular case of the general problem of interpreting a cross between two individuals as if it represents the genetic basis of the difference between two populations.

We appreciate that the reviewer considers our analyses robust and appropriately interpreted. Nevertheless, the reviewer makes very detailed and helpful suggestions, which clearly improve our manuscript, even though the major conclusions are indeed not influenced. We thank the reviewer for these comments. As all of the points raised here are detailed below, we respond to them one by one below.

More specific comments follow by line number:30-32 – this is not a great summary because it starts by contrasting two extremes and then effectively says that there is evidence for some kind of intermediate. It is surprising that Coyne and Orr's book is not cited since it is probably the best summary of the debate up to 2004.

We have revised this section and inserted the citation (lines 30-34).

33 – 'mechanisms' is not a good choice of word here since it implies something that is fabricated by selection for the purpose of isolation

We have replaced “mechanism” with “component” (line 36).

34-36 – also cite Felsenstein 1981. Note that 'ecological divergence' is a component of 'reproductive isolation' and so the wording here is problematic. It should be something like 'ecological divergence and other components of reproductive isolation'. This applies at multiple points in the manuscript.

We have inserted the citation and changed the wording as suggested (lines 37-39).

37 – while ref. 8 is the origin of the term 'magic trait', ref. 13 provided an important discussion of the idea and ref. 6 provided an alternative terminology as well as clarifying the relationship with pleiotropy. If the magic trait/multiple-effect trait idea is central to this manuscript, some of this literature should be cited.

We have changed the citations as requested and introduced a clear distinction between pleiotropy, magic traits and genetic linkage (lines 39-59).

41 – see ref. 6 for a view on the role of pleiotropy. In my view, nothing in the magic trait idea implies a small number of genes: the point is that the trait contributes to multiple components of reproductive isolation and this can be true regardless of its genetic basis. This is part of the reason why confounding the idea with pleiotropy is unhelpful.

We agree with this comment and have inserted a statement that the concept of a magic trait does not include an assumption on the genetic basis of the trait and should not be confused with pleiotropy (lines 44-46).

42-44 – the theoretical basis for this claim should be supported with citations, as well as giving examples in the next sentence.

We have inserted both a citation and examples (lines 51-59).

65 – actually, this is not enough to meet the magic trait definition because there also need to be evidence for 'ecological divergence': the FM and NM types need to be locally adapted to their niches in some sense. Alternatively, hybrids might suffer a fitness cost – a form of incompatibility. This would fit the multiple-effect trait definition advanced by ref. 6 but not the definition of magic traits used here (although I think it might fit the definition originally used by Gavrilets).

We have inserted an explanation of how and why the FM and NM types are expected to be under divergent ecological selection (lines 75-84). Interestingly, it is also true that hybrids are expected to suffer a fitness cost. We have inserted this as a second component of reproductive isolation (lines 84-88).

72 – I think the authors probably mean 'linkage disequilibrium' here, rather than 'genetic linkage'.

We replaced 'genetic linkage' with 'linkage disequilibrium' (line 95).

91-2 – some caution is needed here since F1 and BX hybrids alone do not represent gene flow. A plot of heterozygosity against hybrid index, or an analysis using NewHybrids would help to show whether later generation hybrids occur. However, caution is also needed because of segregation of structural variants that might be shared between ecotypes and can distort Fst and admixture analyses.

We have rephrased the section to be more cautious about the issues the reviewer is raising here (lines 115-117).

98-100 – meaning 'polymorphic in both timing types', I assume. It seems that the data behind the SV calls in figure 1C are not shown anywhere.

We have inserted that inversions are ´polymorphic´ (lines 123-126). The results of SV calling are now given in a new Supplementary File 1 and the sequence data behind it are included in our ENA submission PRJEB54033.

Figure 1 and S4 – running the windowed PCA and the PCAs in Figure 1 without the 4 outlying FM individuals might well give clearer patterns. The rationale behind the grouping into genotypes in Figure 1 is not clear.

We have performed PCA without the 4 outlying FM individuals, but the overall picture did not get clearer. Therefore, we decided to show the full dataset. The rationale behind the grouping into genotypes is explained in the text (lines 132-168).

134-140 – support for I3 is relatively weak, but that is clear from the text.

As the reviewer considers weak support for I3 to be clear from the text, we did not make any changes.

141-6 – this partly answers the question about assignment to genotypes – it uses more than just the PCA shown in Figure 1. At present, this is all quite hard to access, even though I am used to plots of this type.

Yes, we also relied on heterozygosity and admixture analyses. In order to make this clearer, we rewrote the corresponding section of the results (lines 145-149) and inserted a new “Figure 2—figure supplement 2” with the local admixture analysis.

150-2 – it may be worth noting that this helps to explain the LD patterns in Figure S2: because NM segregates only for I1 and I2, there is LD only in the I2 region (the only region where recombination is suppressed). In FM, recombination is suppressed over the length of I1, but we might expect more gene flux in the area of I2 because it is collinear in SI2 individuals. However, this is all without thinking about gene flow between FM and NM. The strong differences in LD seem quite surprising to me, given the previous inference of extensive gene flow.Table S1 gives the margins of the area of SD (with surprisingly high precision). Since LD can extend well beyond breakpoints it may be better not to think of these as estimated breakpoint positions.

We inserted the explanation of the LD blocks based on the distribution of haplotypes (lines 188-192). We also think of the margins of LD as approximations of the inversion breakpoints and state this clearly in Supplementary File 2, as well as in the methods (line 420-423).

158 – again, SV calling data are not shown.

We have inserted the SV calling data and now refer to it in the text (lines 123-126). See comment above.

Figure 3 – it is unfortunate that the 'peaks around day 20' overlap with the 'natural' distributions, and no single cut-off can separate them. Do points around day 25 belong to the 'spurious' distribution? Are conclusions about QTL sensitive to decisions about boundaries?

It is indeed unfortunate that the spurious peaks do partially overlap with the natural distribution of the parental strains. However, QTL analysis solely relies on the phenotype distribution of the F2. Therefore, a single cut-off for all distributions is not required. In the F2 the spurious peak and the actual peak are separated by at least 1 day without any emergence. Hence, the decision about the boundaries is clear and we did not have to test different boundaries in QTL mapping.

Individuals emerging around day 25 indeed belong to the spurious peak. We have adjusted the figure accordingly (now Figure 4).

168 – maybe 'at least four QTL' since it is impossible to exclude the presence of other loci or of more loci under the broad peaks observed (indeed, Figure S8 does suggest multiple loci under some peaks).

We have changed the wording to at least 4 QTL (lines 209-210). We also noticed that CIM in Figure 5—figure supplement 2 (previously Figure S8) suggests multiple loci under some peaks, but these patterns change with size of the exclusion window in CIM, so we decided not to mention them in order not to over-interpret the CIM.

Figure 4A,B – it is not clear why two lines are plotted in 4A for Chr1. The inference that the gene order is 'reversed' in I1 and I2 is relative to the reference genome (and actually quite hard to see in the current plot). Importantly, there is no evidence of recombination suppression in the cross analysed, which implies that the FM and NM parents were homozygous for the same (inverted) arrangement. This is actually quite surprising, given the arrangement frequencies inferred from Figure 1. However, it is unclear whether the methods used would have picked up partial recombination suppression due to heterozygosity in the grandparents (leading to heterozygosity in some but not all of the F1 individuals).

There are two lines for Chr1 because two QTL were detected in Multiple QTL Mapping, and then the LOD profiles for both QTL are plotted along the chromosome. Indeed, the inference that the gene order is reversed is relative to the reference genome.

Regarding recombination suppression, there is partial recombination suppression in Chr1. Note that the genetic distance on the linkage map between markers C1M7 and C1M5 is much smaller on the genetic map compared to the corresponding sequence in the reference genome. We have now determined the inversion genotypes for the parents of the cross to be SI_2_ and I_1_I_2_ (new Figure 5—figure supplement 3) and these genotypes explain the partial recombination suppression in the F1 generation.

185 – does 'largely additive' mean that dominance effects were small or not significant? In Table S4d, the dominance effects look non-significant but there is no comparison of models with or without dominance. Fig, S7B,C have not come through well in my PDF. Here the tests for epistasis seems to exclude dominance.

Indeed, the dominance effects given by the *fitqtl* function (see Supplementary File 5d) are small. The tests for epistasis (*scantwo*) in “Figure 5—figure supplement 1” (previously Supplementary Figur 7) did not produce significant dominance interactions either. In *scantwo* there was a significant interaction for two loci (c1:c2), but in the model selection procedure implemented in *stepwiseqtl* this interaction did not significantly improve the full model and was therefore dropped again from the analysis.

187-90 – this, of course, confirms that other loci are likely to contribute.

Yes, indeed. We included this statement (lines 239-240).

193-5 – see comments above on Figure 4A,B. It needs to be clear what the inferred inversion genotypes of the parents and F1 were in these crosses. This influence interpretation substantially.

The inversion genotypes of the parents were determined to be SI_2_ and I_1_I_2_ (new “Figure 5—figure supplement 3”), so that we can expect partial recombination suppression in the F1. This information was inserted in the Results section (lines 244-246; 253-255).

199 – but these inversions were also apparently not segregating in the cross families.

Here the parental inversion genotypes were SI and II or both SI respectively, again resulting in partial recombination suppression in the F1. We have inserted this information in the Results section (lines 253-255).

207 – Figure 4B does not make this clear because it does not show Fst for each inversion, only for SNPs within inversion. Also note that a new inversion is typically fixed for one allele at any locus where the ancestral arrangement is polymorphic. There are then many combinations of inversion and SNP frequencies in two populations for which Fst is higher for the SNP than for the inversion, without any requirement for gene flux (as suggested on line 209). Gene flux is needed for cases where both arrangements carry both alleles (as seems to happen in some cases in Figure 4D, although the situation here is complicated by the putatively overlapping inversions which can enhance gene flux).

We have calculated the F_ST_ values for inversions 2L, 2R, 3L and 3R and added them to Figure 3 (previously Figure 2). In the text we now refer to both figures 3 and 5 (previously 2 and 4), to make the comparison (lines 263-266). We are aware that a new inversion is usually fixed for one inverted allele, while the ancestral arrangement is polymorphic, allowing for F_ST_ values of polymorphic SNPs within the inversion to exceed the F_ST_ values of the inversion. This implies that individual SNPs within the ancestral arrangement are strongly differentiated between the timing types (while the inverted haplotype is considered to be fixed for one allele, which actually sets some limits to genetic differentiation of the SNP). We have included this possibility in the text (lines 267-269). Our test for the *period* locus then shows that there is not just differentiation in the ancestral arrangement, but gene flux between the arrangements (lines 273-284).

217-8 – there is no expectation that differentiation at a single locus will be protected by an inversion so it would help to reword this sentence a bit. I think the authors mean that differentiation at period is not enhanced by association with other locally-adaptive variants that is maintained by an inversion.

We agree with the reviewer and have reworded the sentence accordingly (lines 281-284).

250-1 – this comes back to issues in the Introduction with the 'magic trait' idea. The basis of this idea is that recombination is less important because recombination cannot separate the two (or more) components of reproductive isolation influenced by the trait. If one component of isolation is local adaptation (as proposed here but not directly established) then each locus influencing the trait will be under divergent selection. An inversion might be favoured because it maintains advantageous combinations of these loci (the Kirkpatrick-Barton mechanism) but this is different from the case where assortative mating (for example) is not under direct selection and so only evolves in response to indirect selection (as in classical models of reinforcement).

We have completely rewritten this paragraph (lines 321-337). Having now introduced a clear distinction between genetic linkage, magic traits and pleiotropy in the introduction, we here argue that given our data we don’t think genetic linkage is the mechanism for coupling the components of reproductive isolation. We elaborate on why recombination is not important for a magic trait and that we expect divergent selection on all loci affecting the magic trait. We conclude that selection on these loci might favor the spread of certain inversion haplotypes and that this creates a new substrate for additional, genetically linked components of reproductive isolation to evolve.

251-3 – this is a slightly odd distinction. I think it needs to be explained a bit more fully and, ideally, related to relevant models for the spread of inversions.

In course of rewriting the whole paragraph (see comment above), we no longer make this distinction. We now explicitly refer on how natural selection might drive the spread of certain inversions (lines 328-333).

265-7 – this comment about what is 'generally assumed' should be supported by citations. It reflects a view in the Introduction that 'magic traits' are expected to have a simple genetic basis, a view that I consider incorrect (see earlier comments).

We agree with the reviewer that magic traits do not need to have a simple genetic basis. Therefore, we have removed the respective section from the discussion.

272-3 – I am not sure that it is helpful to introduce the idea of 'evolutionary branching' here without further discussion. It would be more in line with the rest of the manuscript to say that the timing ecotypes might accumulate other components of reproductive isolation and eventually complete speciation.

We have rewritten the entire paragraph based on comments from the other reviewer. As a consequence, the idea of evolutionary branching is no longer mentioned. The idea that timing ecotypes might accumulate other components of reproductive isolation and eventually complete speciation was added to the paragraph above (lines 328-337).

287-8 – in line with previous comments, I think that describing the genetic basis of a putative ‘magic trait’ is a valuable contribution but does not require any re-evaluation.

We agree with the reviewer and have rewritten the sentence accordingly (lines 378-380).

Methods328-30 – this identifies the margins of the region of LD, which do not necessarily correspond to breakpoints.

We are fully aware that the region of LD can only approximate the breakpoints and have changed the wording accordingly (lines 420-423).

372-3 – this does not provide sufficient information – see comments on the Results.

We inserted a statement of how the peaks are delineated (lines 466-469).

384-5 – this analysis used all F2 families derived from one pair of grandparents, I believe. This should be made explicit. The inversion genotypes of the parents should be determined (and ideally of the F1 individuals, but this would have to be inferred from their offspring, I think), because the parents may have been heterozygous and so the F1 may have varied.

The information that it was a single pair of parents was already given in the section above (lines 463-466). For clarity, we have repeated it in the section on QTL mapping (lines 481-482). We did determine the inversion genotypes of the parents and have inserted the information in the text (lines 244-246; 253-255, see comments above). Indeed, the F1 likely varied for inversion genotypes, leading to recombination suppression in some families, but not others, i.e. partial recombination suppression overall.

389-90 – so far as I can see this was without formal testing for dominance effects.

The R/qtl function we used (*fitqtl*) includes the calculation of dominance effects, which are given in Supplementary File 5d (previously Supplementary Table 4). We wrote that the model we used (y = Q_1_ + Q_2_ + Q_3_ + Q_4_) consisted of “four additive QTL” as a contrast to putative epistatic effects, not as a contrast to dominance effects. In order to avoid the confusion, we have replaced “additive” with “non-epistatic” (lines 489-491).

395-413 – it seems that the results of these analyses are not available anywhere.

Initially, we only picked the support we could find for specific inversions from the large SV dataset. We now include the global results of SV calling (Supplementary File 1) and the sequence data behind it (ENA accession PRJEB54033). We did not include a detailed analysis of all SV calls, because we consider it distracting from the story presented here. A full account of the SV calls will be given in a separate manuscript.

Reviewer #2 (Recommendations for the authors):This study system promises to hold exciting insights into the genetic basis of ecological traits but there are limitations in the analyses and interpretation that temper my overall enthusiasm for the paper. These are included in my public review but I list them with specific analysis suggestions below:1) The QTL mapping relies on a modest number of individuals and there are important details missing. The authors should provide the overall heritability of the trait, which can be estimated from their parental and offspring data with some assumptions. For the estimates of QTL effect sizes from such small samples, there is a common issue known as the Beavis effect that can inflate the effect size of individual QTL. Related to this, the total effect size of the QTL reported is extremely high and the authors should consider (e.g. with simulations) that the effect size is substantially inflated. The authors should report individual effect sizes and explore their power to detect QTL with different effect sizes given their study design. I think that consideration of heritability and overall power will help with interpretation of this result.

We appreciate these comments but we are not sure how they would affect our overall conclusions. Heritability is evidently dependent on the environmental conditions, which would be rather different in nature, compared to the laboratory conditions. It is not clear what we would gain by calculating a heritability for our QTL experiment. Additionally, even if we make the assumption that environmental variance is equal in the F1 and F2 generations and thus calculate the genetic variance as the difference between the variances in the F1 and the F2, we are still lacking an independent estimate of environmental variance and we can still not calculate broad sense heritability. We do not see which additional reasonable assumptions we can make in order to obtain an estimate of heritability. But we provide now the individual effect sizes for the QTLs in the text (they were previously somewhat hidden in the suppl. tables). In addition, we point out that they could be overestimated about 3-fold, due to the Beavis effect (lines 236-240). Given that all evidence points to an oligogenic architecture, an overestimate of effect sizes is actually a conservative assumption.

Similarly, we think that performing simulations to estimate the power of our QTL mapping approach would not help the overall argument (and is also somewhat beyond the scope of the paper). We expect that we are likely missing QTL with minor effects and state so in the revised manuscript (lines 239-240).

2) My major issue with the paper is the interpretation of divergence within the inversion as linked causally to the genes underlying ecological divergence. As the authors observe, divergence will vary within an inverted region. This can be traced to myriad factors, including variation in mutation rate, variation in constraint, patterns of ancestral polymorphism within this region, and variation in gene conversion within the inversion. Given this, I do not think it is valid to interpret the regions of high differentiation as the causal drivers of the ecological differentiation. In my view, the results need to be reframed in the context of ecological adaptation dependent on large inversions and potentially interesting candidate genes, without a claim that the results can be narrowed to individual genes.

We agree that divergence peaks can be traced to many different factors. We were not intending to make the claim that all genes underlying these divergence peaks are identical with the genes underlying ecological divergence. But we do think that the divergent genes are good candidates for being ecologically relevant. In the end, we have tempered our wording – as suggested by the editor and the reviewer – and now speak of “potential candidate genes” which “may” point to specific molecular mechanisms underlying ecological divergence (lines 286-290; 300-306).

We don’t think that we can make the alternative claim suggested by the reviewer, namely that adaptation is dependent on the large inversions. Firstly, the inversions cover the largest part of the genome so that some QTL will necessarily overlap with them. Secondly, except for chr1 there is no differentiation of inversions between ecotypes. We have given the F_ST_ values now in Figure 3 (previously Figure 2) in order to underline there is no differentiation. We have therefore not followed this suggestion.

3) The authors imply in the discussion based on historical results that the ecotype evolved in situ in the last ~60 years. This speculation should be removed as it seems much more likely that the ecotype was either missed in sampling or a recent migrant introduced it to the area in the recent past.

We were not intending to imply that the ecotype necessarily evolved in situ. We have therefore re-written the entire paragraph to include an alternative time-frame (postglacial, i.e. roughly 10,000 years) and to explicitly refer to the possibility of introgression (while we explicitly do not refer to evolution in situ). See lines 363-368.